# Structural Formation of Alkali-Activated Materials Based on Thermally Treated Marl and Na_2_SiO_3_

**DOI:** 10.3390/ma15196576

**Published:** 2022-09-22

**Authors:** Magomed Mintsaev, Sayd-Alvi Murtazaev, Madina Salamanova, Dena Bataev, Magomed Saidumov, Imran Murtazaev, Roman Fediuk

**Affiliations:** 1Grozny State Oil Technical University Named after Acad. M.D. Millionshchikov, 364051 Grozny, Russia; 2Kh. Ibragimov Complex Institute of the Russian Academy of Sciences, 364051 Grozny, Russia; 3Polytechnical Institute, Far Eastern Federal University, 690922 Vladivostok, Russia; 4Peter the Great St. Petersburg Polytechnic University, 195251 St. Petersburg, Russia

**Keywords:** silicified marl, heat treatment, alkaline activation, liquid glass, microstructure

## Abstract

Modern materials science is aimed towards abandoning Portland cement in the production of building materials. The scientific novelty of this study lies in its being the first time a comprehensive study of the structure formation of alkali-activated materials (AAM) based on thermally treated marl and Na_2_SiO_3_ is carried out. The tasks for achieving this goal were to characterize the thermally treated marl as a new binder, and to comprehensively research the microstructure, fresh, physical, and mechanical properties of the AAM based on the binder. Received active particles of marl with a smaller size than Portland cement have a specific surface area of 580–590 m^2^/kg. The mineral composition of heat-treated marl is characterized by calcium silicates, which guarantee good binding properties. The results of X-ray diffraction analysis of the samples based on the clinker-free binder of alkaline activation using opoka-like marl confirmed the presence of calcite, quartz, and feldspar close to albite, micas, and zeolites. The obtained products of the chemical interaction of the components of the binder confirm the effectiveness of the newly developed AAM. As a result of comparing several binders, it was found that the binder “thermally treated marl—Na_2_SiO_3_” is the most effective, since for specimens based on it, a maximum compressive strength of 42.6 MPa, a flexural strength of 4.6 MPa, and minimum setting time were obtained (start 26 min, end 32 min) as well as a water absorption of 10.2 wt.%. The research results will be of interest to specialists in the construction industry, since the proposed recipes for clinker-free cements are an alternative to expensive and energy-intensive Portland cement and provide the creation of strong and durable concrete and reinforced concrete composites.

## 1. Introduction

Analysis of the modern construction market has shown that Portland cement (PC) has been considered the most highly demanded material in construction for many years [1]. However, at the same time, PC production belongs to the category of harmful and environmentally unfavorable materials, which is justified by the large-scale consumption of natural resources; it is also potentially hazardous to health due to its emissions in the form of reactionary aspiration and clinker dust, lime, large volumes of carbon dioxide and sulfur compounds, dioxins, and heavy metals, etc., [2]. On the agenda of many environmental forums is the issue of the decarbonization of the economy, due to the reduction of greenhouse gas emissions, which cause a warming of the planet’s climate [3].

As it is known, the evolution of the development of modern concrete science has confirmed the effectiveness of the multicomponent systems development using two or more highly dispersed mineral powders of various natures and high-quality fractionated coarse and fine aggregates [4]. Despite the resource intensity and high cost of Portland cement, traditional building composites based on it still dominate in recent years. Environmental issues are drawing the attention of the scientific community to the search for new materials on clinker-free, low-clinker, or filled binder systems using highly effective hyper- and superplasticizers [1,5,6].

The clinker-free technology of alkaline mixing binders has found recognition and industrial applications in many developed countries of the world, and all this is due to its improved fresh and hardened properties and performances [7]. The origins of this technology come from Europe, where Emil Langin invented slag alkali cement in 1862 [8]. The obtained material proved itself well and occupied its niche in the construction market [9]. Slag alkali cement was widely used throughout Europe, and competed with Portland cement [10]. It was used in the construction of the subway in Paris, widely used in Germany, and at the same time, its industrial uses began in the United States [11]. The European standard for slag alkali cement was adopted at the beginning of the 20th century [12,13].

Alkali-activated materials (AAM) are very promising and are constantly being newly used all over the world [14]. Countries such as the UK, Japan, Singapore, and the Netherlands are leaders in the production of this binding material [15]. In addition, manufacturers of this material reduce the negative environmental impacts [16]. This solves the problem of energy-intensive high-temperature firing and eliminates the addition carbon dioxide emissions into the atmosphere [17]. Technical characteristics of AAM are unique in comparison with Portland cement materials: it possesses high strength characteristics, chemical resistance, good workability, resistance to chlorides, low heat evolution, and short setting time; and all these things combined will determine the scope of the application of the slag alkali cement in construction [18,19,20].

Zhang et al. [21] studied the compressive strength and anti-chloride ion penetration assessment of geopolymer mortar merging PVA fiber and nano-SiO_2_. Han et al. [22] researched comprehensive reviews of the properties of fly ash-based geopolymer with the additive of nano-SiO_2_. Wang et al. [23] investigated the effect of municipal solid waste incineration ash on the microstructure and hydration mechanisms of geopolymer composites.

The weak development of this type of material in most other countries is associated with a number of problems: the limited and unstable raw material base, the lack of a sufficient regulatory framework, a shortage of alkaline solutions, and poor information content of the products [24,25].

There are scientific developments [26,27] and experience in the practical implementation of this technology. Currently, the production of clinker-free alkaline binders and concrete based on them is becoming more than relevant, since there exist millions of tons of waste of ferrous metallurgy in the form of granulated blast furnace slag [28,29]. Chemical industry waste from the production of alumina, sodium sulfides, and caprolactam are also essentially not used [30]. In addition, there is a significant problem in their recycling [31]. All these factors can contribute to the development of clinker-free technology and reduce the cost of obtaining binders by 50% [32,33]. It is worthy of note that for the preparation of alkaline concrete, it is possible to use fine and coarse aggregate without restrictions on the content of clay and dust fractions, since these particles enter into chemical interaction with alkali metal compounds, forming sodium hydroaluminosilicates [34]. Thus, the cost of obtaining building composites will significantly decrease without any detriment to their strength characteristics [35,36].

Clinker-free technology of alkaline binders is promising and quite effective; however, in many countries, there are many regions in which industrial wastes of the metallurgical industry are deficient raw materials and their use is economically unprofitable due to transportation costs [37]. Considering the fact that blast furnace slags are characterized by aluminosilicate composition, and the mineral and chemical compositions are not stable, due to changes in the composition of the fuel, the technological process of smelting and storage, etc., it is necessary to find an alternative to them [38].

The use of highly dispersed powders of aluminosilicate and siliceous genesis for subsequent mixing with an alkaline solution will make it possible to synthesize geopolymer composites that are not inferior to slag-alkaline concretes [39]. The novelty of the research consists in the development and addition of the theoretical foundations of AAM structure formation obtained by alkaline activation of thermally treated silicified marl.

The aim of this work is a comprehensive study of the structural formation of alkaline-activated materials based on thermally treated marl and Na_2_SiO_3_. The tasks to achieve this aim were the characterization of the thermally treated marl as a new binder and a comprehensive study of the microstructural, fresh, physical, and mechanical properties of alkali-activated material based on the binder.

## 2. Materials and Methods

### 2.1. Materials

Natural quarry-silicified marl (North Caucasus, Russia) was used as the basis of a cementless binder (precursor). Table 1 details chemical composition of the marl. The specific surface area of the marl is 580 m^2^/kg. Thermal treating of the marl was carried out in a muffle furnace at a temperature of 700 °C for 1 h. The specific surface area of the mineral powders prepared for alkaline activation was 580–590 m^2^/kg, the predominant particle size ranged from 1 to 5 µm, and the true density was 2.6 g/cm^3^.

Figure 1 presents the appearance of the initial and treated marl determinated by XRF.

Silicified marl (opoka) consists of opal with impurities of clay minerals, mineral grains, and skeletons of microorganisms. It is assumed that the simultaneous presence of calcite and silica will favorably affect the properties of the multicomponent system, but heat treatment at a temperature of 700 °C will increase the pozzolanic activity of this mineral additive when interacting with calcium hydroxide and water.

Liquid glass, sodium metasilicate Na_2_SiO_3_, was used as the binder modifier (silicate modulus 2.8, density 1.42 g/cm^3^). Sodium hydroxide, NaOH, was used as an alkali. To accelerate the hardening process of AAM, sodium fluorosilicon Na_2_SiF_6_ was used in a dosage of 6% of the Na_2_SiO_3_ mass.

The quartz sand obtained by fractionation of fine (1.5 mm) and coarse (2.5 mm) grains in a ratio of 22%:78% was used as a fine aggregate.

### 2.2. Mix Design

Four different concrete mixes were developed; common for all was the use of marl (Table 2). Two mixes were with initial marl (M1 modified with Na_2_SiO_3_ and M2 with Na_2_SiO_3_ and hardening accelerator Na_2_SiF_6_). The other two mixes were with marl thermally treated at 700 °C (TM1 modified with Na_2_SiO_3_ and TM2 mixed with water to prove the effectiveness of the precursor).

The prepared samples hardened on the first day under normal conditions at a temperature of 20 ± 2 °C, but since the second day the samples were placed periodically, for 28 days, in an oven at a temperature of 50 °C for two hours.

### 2.3. Methods

Granulometry of the particles of the raw materials was carried out using a laser analyzer, Analysette 22 (Fritsch, Idar-Oberstein, Germany). The specific surface of bulk raw materials was studied using the PSH-12 device (Khodakov Devices, Moscow, Russia).

Silicified marl was studied in two forms: before heat treatment in its natural form, and after thermal treatment at a temperature of 700 °C. X-ray fluorescence (XRF) analysis of the marl was carried out by a universal X-ray spectrometer Clever C-31 (Eleron, Elektrostal, Russia).

Studies of the macro- and microstructure of mineral raw materials and studies of energy-dispersive microanalysis were carried out using a Quanta 3D 200i scanning electron microscope (FEI Company, Hillsboro, OR, USA) with an integrated Genesis Apex 2 EDS microanalysis system (EDAX, Mahwah, NJ, USA). The obtained spectra were processed using the EDAX TEAM software V1.1.55 resource (EDAX, Mahwah, NJ, USA).

The processes’ structural formations of the AAM were studied using a scanning electron microscope (SEM) Vega II LMU (Tescan, Brno, Czech) with an energy dispersive microanalysis system Inca energy 450/XT (Silicon Drift detector (ADD; resolution 133 eV at a count rate of 20,000 pulses/s) manufactured by the company Oxford Instruments Analytical (Oxford, UK). The system provides the ability to carry out elemental analysis in the range from Na to U (lighter elements are not determined; oxygen is calculated by stoichiometry). The studies were carried out at an accelerating voltage of 20 kV.

X-ray diffraction (XRD) analysis was performed for reflection according to Bragg–Brentano by an ARLX’TRA diffractometer using the Θ–Θ kinematic scheme with a horizontal arrangement of a flat sample. The characteristic radiation of a copper anode was used (wavelengths CuKα1 1.5406 Å, CuKα2 1.5444 Å). The energy window of the semiconductor detector tuned to register this range also partially captures close wavelengths of CuKβ1 1.3922 Å and WLα1 1.4763 Å.

Cement paste samples were studied by an IR Prestige-21 IR-Fourier spectrometer (Shimadzu, Kyoto, Japan) with a Miracle frustrated total internal reflection attachment (PikeTechnologies, Madison, WI, USA) to perform Fourier-transform infrared spectroscopy (FTIR).

Studies of the fresh and hardened properties of AAM were carried out in accordance with Russian standard 30744-2001. Compressive strength was determined on 70 × 70 × 70 mm samples, but theflexural strength tests used samples-beams with a size of 40 × 40 × 160 mm (three samples of each composition).

## 3. Results and Discussion

### 3.1. Characterization of the Thermally Treated Marl

Opoka-like marl particles were studied by scanning-electron microscopy and X-ray diffraction analysis before and after thermal exposure at a temperature of 700 °C (Figure 2, Table 2 and Table 3), and the results confirmed the similar nature of the microparticles’ structures, in both cases lamellar (particle size 1–5 µm). Therefore, they are much smaller then particles of Portland cement. Visual inspection of the SEM images did not reveal a clear difference in the microstructure. The only difference is the more loose and porous relief of the surface of the marl in its natural state (Figure 2a,b). The grain structure after the thermal treating is characterized by a thinly-crystalline non-uniformly distributed structure (Figure 2c,d) with round closed pores.

The natural phase spectra of the silicified marl bulk revealed the predominant presence of minerals such as calcite, quartz, and aluminosilicates, represented by feldspars and kaolinite (Figure 3).

Phase spectra of the silicified marl bulk after thermal treating at 700 °C revealed the presence of larnite Ca_2_SiO_4_ (Figure 4).

The results of X-ray diffraction analysis (Figure 5) confirmed the presence in the thermally treated opoka-like marl samples of the peaks belonging to calcite, quartz, and a very small number of feldspars, most likely plagioclases. The peak at ~9 degreescorresponds to micas or hydromicas.

Reflexes similar to those of dibasic calcium silicates, such as larnite and calcium oxide, were found. The bright peak overlaps with one of the lines of the putative larnite; the second one is not bright and coincides with the halo-like area, possibly associated with the presence of weakly crystallized calcium silicates. Moreover, there is a small halo that can be identified as the portlandite phase in the region of 18 degrees.

X-ray diffraction analysis established the presence of calcium silicates and aluminosilicates of various basicity. Therefore, at the next stage, the goal was to establish the effectiveness of the selected processing technique, since heat treatment entails certain costs for energy and equipment.

### 3.2. Microstructure of Alkali-Activated Material

Prepared silicified marl powders subjected to both fine grinding and heat treatment were mixed with water and an alkaline solution to solve the aforementioned problem. The results of the electron probe analysis of the studied samples showed that the structure of the hardened sample is characterized by an inhomogeneous fine-grained structure (Figure 6c) and an aggregative structure (Figure 6b). In the bulk of the structure, clusters up to 100–200 µm in size were found, in some places framed by films of amorphous sodium silicate hydrates, which are distinguished by a rather dense composition and an increased content of calcite (Figure 6e). The obtained results of the study of the developed composite microstructure confirm a similar geopolymerization mechanism, for example, in comparison with metakaolin/fly ash-based geopolymer [40].

The analysis results of the studied areas confirmed that the microstructure of the groundmass is formed by non-crystallized aggregates of hydroaluminosilicate “zeolite” composition with a variable Ca/Na ratio (Figure 7, Table 3, spectras 1 and 2), calcite (Table 3, spectras 3, 5, 6), phases similar in composition to dicalcium silicate hydrates (Table 3, spectra 4), and, possibly, calcium hydroxide. Iron and magnesium are associated with aluminosilicate hydrated compounds.

**Table 3 materials-15-06576-t003:** Results of typical microphases analysis (spots of analysis are in the Figure 7).

Spectra	Na_2_O	MgO	Al_2_O_3_	SiO_2_	K_2_O	CaO	Loss of Ignition
**1**	11.46	0.00	19.49	54.26	0.00	2.34	12.46
**2**	6.29	0.00	15.72	57.93	0.00	9.01	11.05
**3**	1.14	0.23	0.29	2.19	0.00	41.97	54.19
**4**	2.23	0.52	0.91	16.70	0.18	39.59	39.87
**5**	0.85	0.55	0.56	6.69	0.00	42.13	49.22
**6**	0.82	0.00	0.56	4.97	0.00	39.96	53.69

The results of X-ray diffraction analysis of the samples based on the clinker-free binder of alkaline activation using opoka-like marl confirmed the presence of calcite, quartz, feldspar close to albite, micas, and zeolites (Figure 8). Potassium feldspar was found in the fine aggregate.

It was found that as a result of the destruction of the alumino-silicon-oxygen framework and bonding with alkali metal oxides, the synthesis of the hydroaluminosilicate zeolite phase Mn + x/n[(AlO_2_) − x(SiO_2_)]zH_2_O of variable composition occurs, which contributes to the creation of concrete and mortar composites with improved physical, mechanical, and technical-economic indicators.

Consequently, the obtained products of the chemical interaction of the components of the binder confirm the effectiveness of the new developed AAM. The phase composition of new formations and industrial experience in the use of slag-alkaline concretes guarantee its success in such special segments of construction, where urgent repair work, fast setting of the mixture, corrosion resistance, reduced exotherm in the production of large-sized products and structures, and frost resistance are required.

Prepared cement paste samples of based on the binder “thermal marl (700 °C)—alkaline solution” were subjected to FTIR, based on the selective absorption of the infrared part of the spectrum by a substance when this radiation passes through it. On FTIR pattern, minerals of the calcite and dolomite groups are characterized by absorption peaks at 1450–1435 cm^−1^, 887–897 cm^−1^, and 748–710 cm^−1^ (Figure 9).

Thus, from the physicochemical analyses of hydration products, the consumption of building materials is higher than that of the composites on clinker-free binders of alkaline activation using substandard secondary and chemical raw materials. There are formations of minerals such as: quartz SiO_2_ with d/n (4.24; 3.34; 2.45; 2.28; 2.23; 2.12; 1.81; 1.53Å); calcite CaCO_3_ with d/n (3.84; 3.029; 2.49; 2.277; 1.91; 1.86; 1.52 Å); orthoclase K_2_O · Al_2_O_3_ · 6SiO_2_ with d/n (6.44; 4.25; 4.02; 3.18; 2.99; 2.28; 1.72; 1.53 Å).

The obtained results prove similar mechanisms of structure formation and geopolymerization in comparison with the previously studied metakaolin-fly ash blend alkali-activated sustainable mortar [41].

### 3.3. Fresh, Physical and Mechanical Properties of Alkali-Activated Material

The standard consistency of the AAM based on the binder “thermally treated marl—Na_2_SiO_3_” is characterized by a high demand for an alkaline solution of 56%. The setting time is rather short, beginning at 26, ending at 32 min. Even when mixing water, the powder of thermally activated marl exhibits astringent properties; setting occurred in 97 min, activity 6.7 MPa (Table 4). It is the use of such technological methods as thermal and alkaline activation that made it possible to achieve high performance. Fresh properties of the developed alkali-activated materials correspond to similar characteristics of geopolymer composites prepared from previously studied precursors [42].

Thus, the binder “thermally treated marl—Na_2_SiO_3_” is the most effective, since the maximum compressive strength of 42.6 MPa was obtained for samples based on this binder. As a result, the formation of the AAM structure was studied precisely on these samples. Previously established patterns of the processes of formation of the structure and properties of alkali-activated materials based on granulated blast-furnace slag [6,7,15,16,17] and carried out in the framework of the research of the structure of AAM with mineral powder from thermally treated marl showed that the structural formation algorithm is largely similar for both.

The formation of the structure of alkaline-activated material, for which almost all components are the active components, can be divided into the following phases:

I phase—characterized by an increase in the pH of the medium by cations of alkali and alkaline earth metals, which leads to the destruction of the aluminum-silicon-oxygen skeleton, and as a result of the cation exchange 2Na^+^ ↔ Ca^2+^, hydroaluminosilica of a variable nature is formed.

II phase—the synthesized condensation structure is transformed into a solid phase; the process of crystallization of the structure begins. The cation exchange 2Na^+^ ↔ Ca^2+^ continues and promotes the binding of silicates and aluminosilicates by alkali metal cations.

III phase—the formed initial solid phases are characterized by an unstable character and, as a result of growth and development, are transformed into a strong crystalline intergrowth.

AAM paste can be characterized by the following structural elements:-dispersion medium;-diffusion interfacial transition zone “powder particles—gel Na_2_SiO_3_”;-reactive mineral powder consisting of reacted and unreacted parts [8,17].

As a result of activating of the mixture of mineral powder and aggregate with an alkaline solution, the energy potential on the surface of the particles of the mineral powder increases, the space between the grains of the solid phase decreases, the liquid phase is redistributed toward the contact zone “powder particles—gel Na_2_SiO_3_”, and the surface of the mineral particles becomes the substrate for the crystallization of new growths.

Thus, the proposed clinker-free technology for obtaining binders of alkaline activation using marl thermally treated at a temperature of 700 °C will reduce the load on the natural potential and ecological safety of the environment. Moreover, given the shortage of ferrous metallurgy waste in many regions of the world the proposed technology will make it possible to obtain binders grades M300–400, eliminating the huge transport costs for the transportation of granulated blast furnace slags and additional costs for mechanical activation in expensive grinding equipment. The results obtained confirm the effectiveness of the clinker-free technology, since the obtained new building materials are guaranteed to ensure successful industrial implementation in such construction sites where urgent repair work is necessary, due to the rapid setting of the mixture, corrosion resistance, and reduced exotherm for the production of large-sized products.

Further development of the research topic may be associated with the development of new constructive technological solutions for the repair and restoration of structures of buildings and structures, which make it possible to reduce the cost of repair work of concrete and reinforced concrete elements and to expand and improve the formulations and technology for obtaining clinker-free binders on substandard and technogenic raw materials.

## 4. Conclusions

A comprehensive study of the structural formations of alkali-activated materials based on thermally treated marl and Na_2_SiO_3_ was carried out. The tasks for achieving this goal were to characterize the thermally treated marl as a new binder and to comprehensively research the microstructural, fresh, physical, and mechanical properties of the alkali-activated material based on the binder. The following main conclusions were made, emphasizing the scientific novelty and practical significance of the work.

1. Particles of the active marl with a smaller size than Portland cement were received. The mineral composition of heat-treated marl is characterized by calcium silicates, which guarantee good binding properties.

2. The results of XRD, FTIR, and SEM analyses of the samples based on the clinker-free binder of alkaline activation using opoka-like marl confirmed the presence of calcite, quartz, feldspar close to albite, micas, and zeolites. The obtained products of the chemical interaction of the binder components confirm the effectiveness of the new developed AAM.

3. As a result of comparing several binders, it was found that the binder “thermally treated marl—Na_2_SiO_3_” is the most effective, since for specimens based on it, a maximum compressive strength of 42.6 MPa, a flexural strength of 4.6 MPa, and minimum setting time were obtained (start 26 min, end 32 min) and water absorption 10.2 wt.%.

4. The research results will be of interest to specialists in the construction industry, since the proposed recipes for clinker-free cements are alternatives to expensive and energy-intensive Portland cement, and assist in the creation of strong and durable concrete and reinforced concrete composites.

## Figures and Tables

**Figure 1 materials-15-06576-f001:**
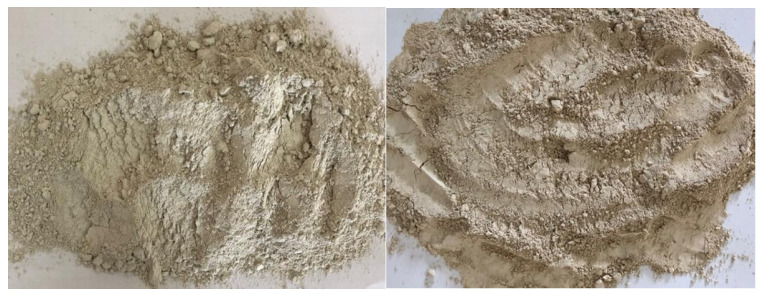
Appearance of the initial and treated marl.

**Figure 2 materials-15-06576-f002:**
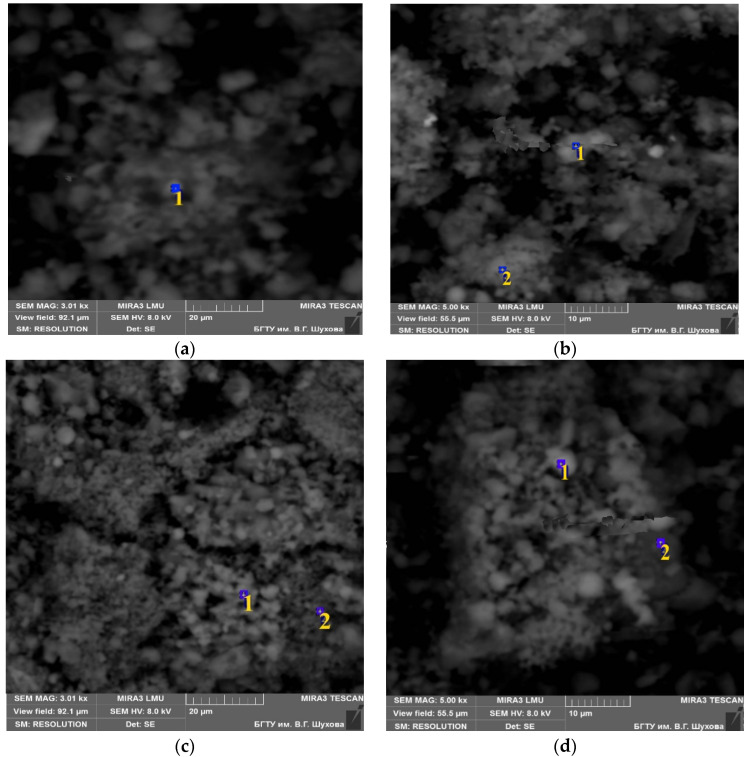
SEM images of initial silicified marl without treatment (**a**,**b**) and thermally treated at 700 °C (**c**,**d**). 1 and 2—research EDS points in Figure 3 and Figure 4.

**Figure 3 materials-15-06576-f003:**
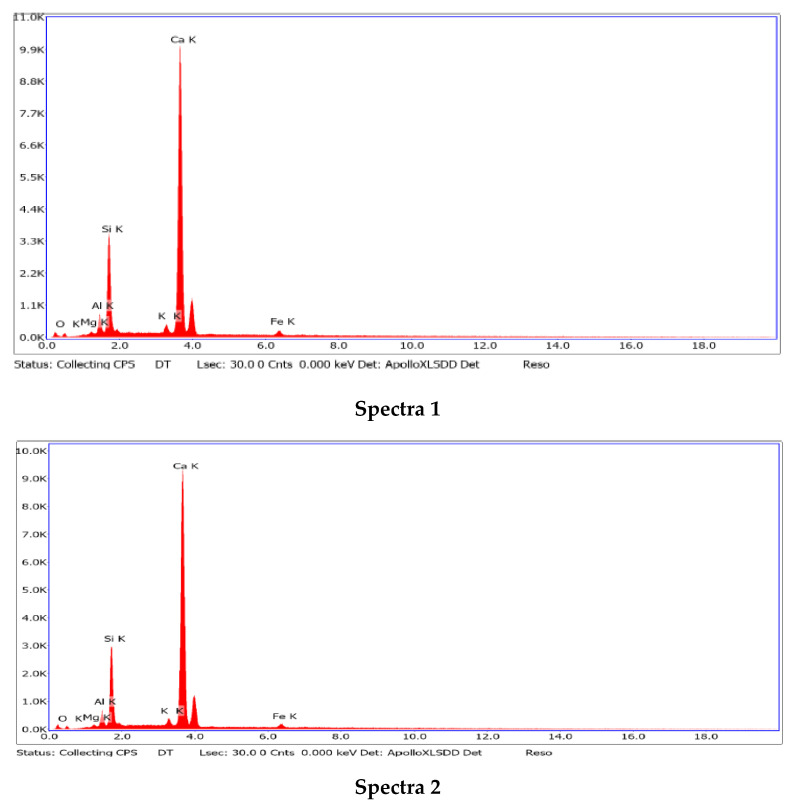
EDS spectra phases of the initial marl.

**Figure 4 materials-15-06576-f004:**
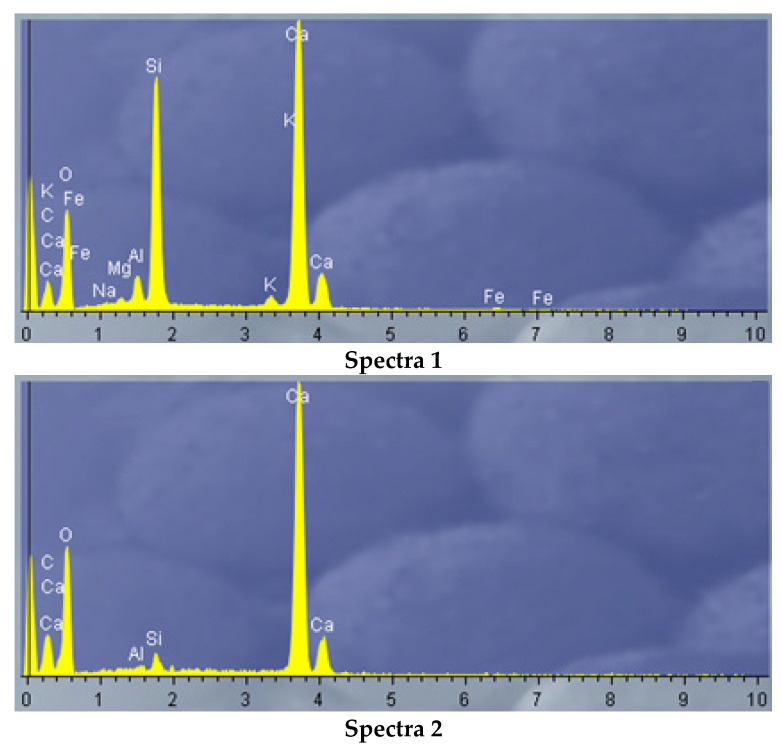
EDS spectra phases of the marl after thermal treating.

**Figure 5 materials-15-06576-f005:**
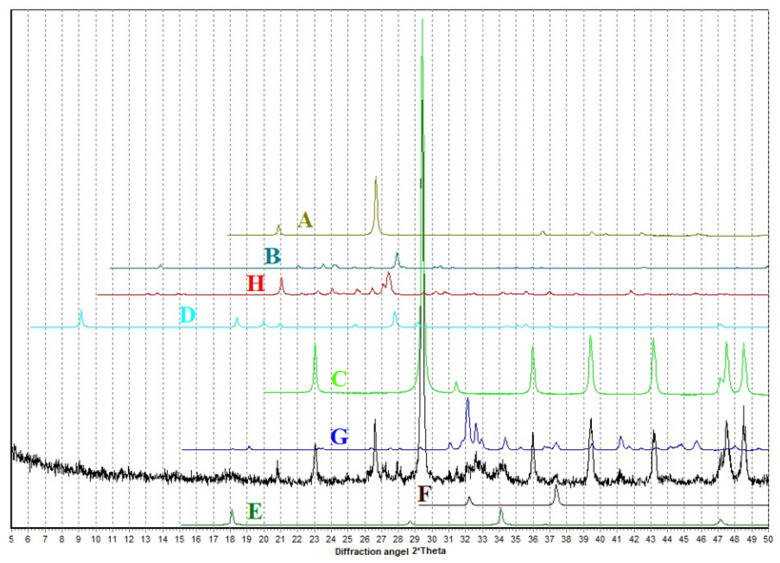
Diffraction pattern of thermally treated (700 °C) marl sample, in comparison with the database PDF-2. The comparison phases are as follows: A—quartz, B—calcite, C—albite, D—paragonite, E—portlandite, F—calcium oxide, G—larnite, H—microcline/orthoclase.

**Figure 6 materials-15-06576-f006:**
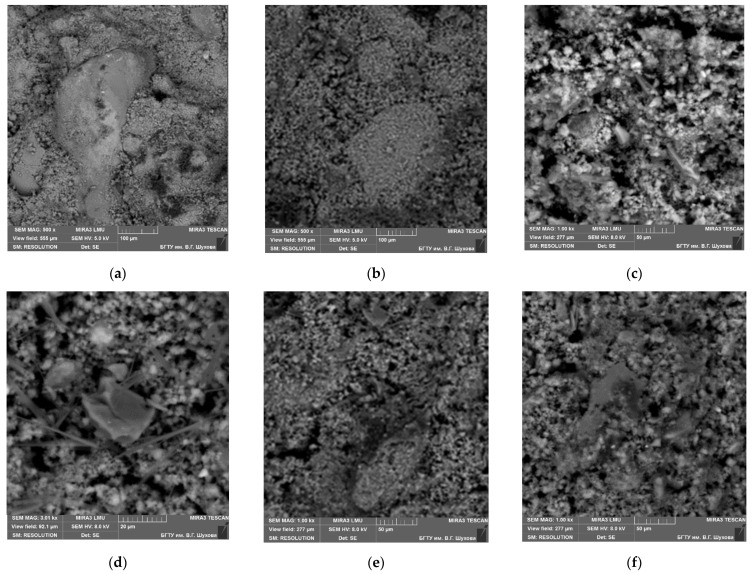
Microstructures of AAM with thermally treated marl (sample TM1). (**a**) Sample material structure. (**b**) Clusters of essentially calcite composition. (**c**) Typical sample microstructure. (**d**) Sodium silicate hydrates. (**e**) Microcrystalline cluster coated with a film of sodium hydrosilicate. (**f**) Sodium silicate hydrates in the structure of the material.

**Figure 7 materials-15-06576-f007:**
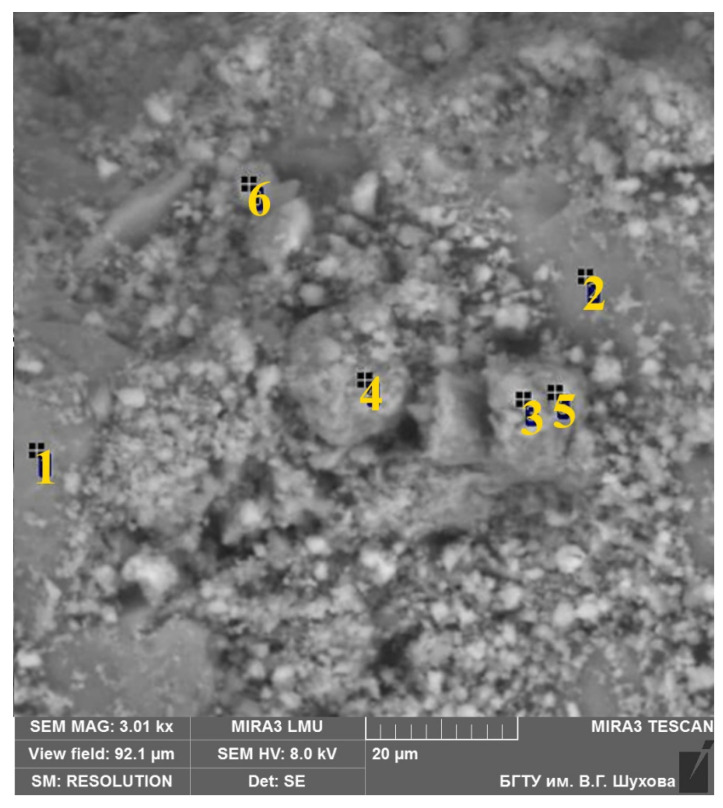
Typical microstructure of the AAM (sample TM1). 1–6—spots of analysis from Table 3.

**Figure 8 materials-15-06576-f008:**
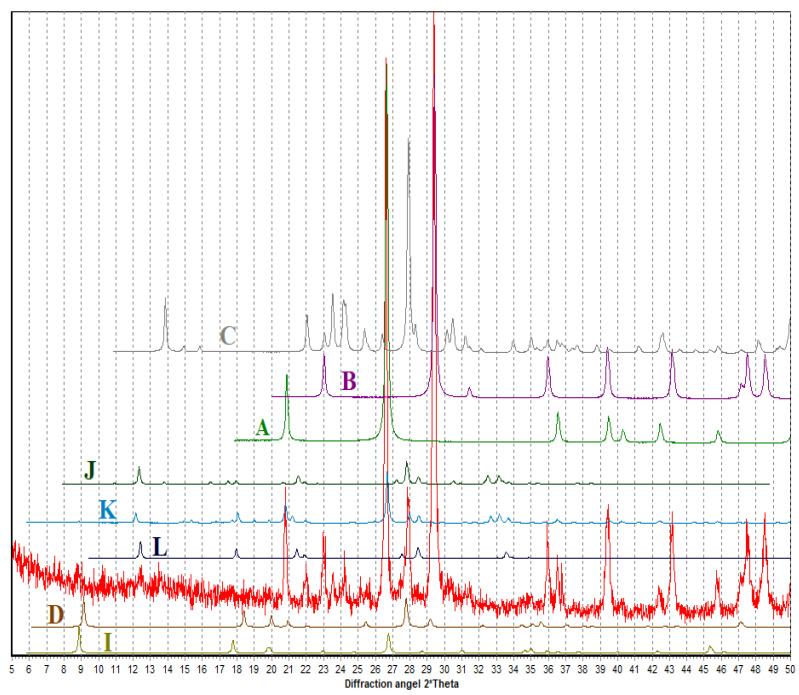
Diffraction patterns of the AAM based on thermally treated (700 °C) marl (sample TM1), in comparison with the database PDF-2. The comparison phases are as follows: A—quartz, B—calcite, C—albite, D—paragonite, I—muscovite, J—phillipsite, K—gismondite, L—garronite.

**Figure 9 materials-15-06576-f009:**
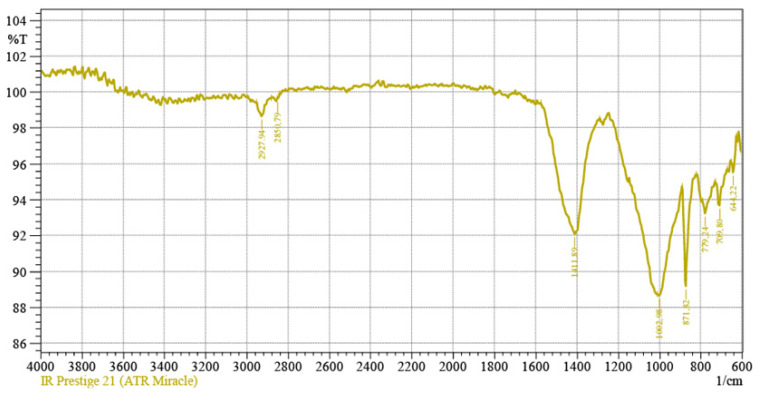
FTIR pattern of the cement paste based on the binder “thermally treated marl (700 °C)—alkaline solution” (sample TM1).

**Table 1 materials-15-06576-t001:** Chemical composition of the raw, wt.%.

Marl	CaO	SiO_2_	Al_2_O_3_	Fe_2_O_3_	Na_2_O	MgO	CO_2_	Loss on Ignition
Initial	66.61	12.29	2.23	1.16	0.38	0.25	17.08	-
Treated	61.53	12.11	2.07	1.12	0.29	0.10	-	22.78

**Table 2 materials-15-06576-t002:** Mix proportions.

Mix ID	Components, kg per 1 m^3^						
	Marl	Marl Thermally Treated at 700 °C	Na_2_SiO_3_	H_2_O	Na_2_SiF_6_	NaOH	Sand
M1	700	-	280	-	-	70	1040
M2	700	-	280	-	16.8	70	1040
TM1	-	700	280	-	-	70	1040
TM2	-	700	-	280	-	70	1040

**Table 4 materials-15-06576-t004:** Fresh, physical, and mechanical properties of AAM.

Properties	TM1	TM2	M1	M2
**Normal density of AAM, %**	56.0	40.0	51.0	52.0
**Setting time**	00–26	01–37	01–07	00–55
**Start/end, hours-min**	00–32	06–29	02–29	01–43
**Average density, g/cm^3^**	1.90	1.80	2.00	2.01
**Water absorption, wt.%**	10.2	11.4	11.9	11.7
**Strength, MPa:**				
**Flexural**	4.7	0.2	1.0	1.1
**Compressive**	42.6	6.7	9.0	9.6

## Data Availability

Not applicable.

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
