# Peer review of "Structural Formation of Alkali-Activated Materials Based on Thermally Treated Marl and Na2SiO3"

_materials, 2022, doi:10.3390/ma15196576_

Round 1
Reviewer 1 Report
For the first time, a comprehensive study of the structure formation of alkali-activated materials (AAM) based on thermally treated mar and Na2SiO3 was carried out in this study. This is interesting and novelty. There are minor errors to be corrected before accepted for publication.
1. The research background should be improved. For example, i do not entirely agree with this sentence "Traditional building composites are losing their importance 48 due to the resource intensity and high cost of Portland cement". Traditional building composites still dominate in recent years.
2. The graphic content is not clear enough. For example, Figs. 2, 6 and 7.
Author Response
Dear Reviewer 1!
Thank you for your interest in our manuscript. Your valuable comments helped make the manuscript even better. All corrections in the manuscript are highlighted in blue.
Comment 1: The research background should be improved. For example, i do not entirely agree with this sentence "Traditional building composites are losing their importance 48 due to the resource intensity and high cost of Portland cement". Traditional building composites still dominate in recent years.
Response: The research background has been carefully improved.
Comment 2: The graphic content is not clear enough. For example, Figs. 2, 6 and 7.
Response: The graphic content has been carefully improved.
Reviewer 2 Report
The work is well performed and presented. however, it is important to mention that author conclusions are only based on XRD, SEM and EDS, however these characterizations are not sufficient. I suggest authors to add other spectroscopies like FTIR or Raman in order to confirm the functional groups and vibrational modes and can come to the conclusions by correlating with the XRD, and SEM.
Author Response
Dear Reviewer 2!
Thank you for your interest in our manuscript. Your valuable comments helped make the manuscript even better. All corrections in the manuscript are highlighted in blue.
Comment 1: The work is well performed and presented. however, it is important to mention that author conclusions are only based on XRD, SEM and EDS, however these characterizations are not sufficient. I suggest authors to add other spectroscopies like FTIR or Raman in order to confirm the functional groups and vibrational modes and can come to the conclusions by correlating with the XRD, and SEM.
Response: Fig. 9 has been added with the results of the FTIR, as well as 2 paragraphs describing them
Reviewer 3 Report
This paper focuses on the fabrication of alkali-activated materials (AAM) based on thermally treated marl and Na2SiO3. Also, the microstructure, fresh, physical and mechanical properties of the new construction material were investigated. This work has important guidance values for the further research and application of alkali-activated materials. This research is very interesting and provides a significant contribution to the knowledge of this field. Therefore, in my opinion, it can be accepted after some major revision. Several comments are given below:
(1) In the abstract, it is stated that “As a result of comparing several binders, it was found that the binder "thermally treated marl - Na2SiO3" is the most effective,…”. The comparing results should be added.
(2) There are too many keywords. The keywords of clinker - free binders, zeolite phase, aluminosilicates can be removed.
(3) There are a lot of latest studies conducted on alkali-activated materials. So the related research progress should be introduced in the section of Introduction, and some references can be used: Compressive strength and anti-chloride ion penetration assessment of geopolymer mortar merging PVA fiber and nano-SiO2 using RBF–BP composite neural network, Nanotechnology Reviews, 2022; Comprehensive review of the properties of fly ash-based geopolymer with additive of nano-SiO2, Nanotechnology Reviews, 2022; Effect of municipal solid waste incineration fly ash on the mechanical properties and microstructure of geopolymer concrete, Gels, 2022.
(4) At the end of the section of 2.1, the expression of the ratio of fine and coarse quartz sand “22:78%” is wrong. Please revised it.
(5) The detailed information on thermal treating of marl should be provided.
(6) How about the grain size of the activated marl?
(7) The quality of the SEM images in Figures 2, 6, and 7 should be improved.
(8) In most of the paragraphs for results and discussion, the analysis and discussion should be strengthened, especially in the mechanism analysis. Some contrastive analysis between this study and other related literature should be added, and some references can be used: High-temperature behavior of polyvinyl alcohol fiber-reinforced metakaolin/fly ash-based geopolymer mortar, Composites Part B - Engineering, 2022; Influencing factors analysis and optimized prediction model for rheology and flowability of nano-SiO2 and PVA fiber reinforced alkali-activated composites, Journal of Cleaner Production, 2022; Influence of SiO2 /Na2O molar ratio on mechanical properties and durability of metakaolin-fly ash blend alkali-activated sustainable mortar incorporating manufactured sand, Journal of Materials Research and Technology, 2022.
(9) The section of 4. Conclusion should be revised. The main conclusion of this study should be presented in three or four pieces.
Author Response
Dear Reviewer 3!
Thank you for your interest in our manuscript. Your valuable comments helped make the manuscript even better. All corrections in the manuscript are highlighted in blue.
Comment 1: In the abstract, it is stated that “As a result of comparing several binders, it was found that the binder "thermally treated marl - Na2SiO3" is the most effective,…”. The comparing results should be added.
Response: This comparison is given in Table 4
Comment 2: There are too many keywords. The keywords of clinker - free binders, zeolite phase, aluminosilicates can be removed.
Response: Removed
Comment 3: There are a lot of latest studies conducted on alkali-activated materials. So the related research progress should be introduced in the section of Introduction, and some references can be used: Compressive strength and anti-chloride ion penetration assessment of geopolymer mortar merging PVA fiber and nano-SiO2 using RBF–BP composite neural network, Nanotechnology Reviews, 2022; Comprehensive review of the properties of fly ash-based geopolymer with additive of nano-SiO2, Nanotechnology Reviews, 2022; Effect of municipal solid waste incineration fly ash on the mechanical properties and microstructure of geopolymer concrete, Gels, 2022.
Response: A paragraph has been added to the introduction devoted to a detailed analysis of the articles kindly suggested by you
Comment 4: At the end of the section of 2.1, the expression of the ratio of fine and coarse quartz sand “22:78%” is wrong. Please revised it.
Response: Revised
Comment 5: The detailed information on thermal treating of marl should be provided.
Response: Thermal activation of the marl was carried out in a muffle furnace at a temperature of 700°C for 1 hour.
Comment 6: How about the grain size of the activated marl?
Response: Added: «predominant particle size from 1 to 5 µm»
Comment 7: The quality of the SEM images in Figures 2, 6, and 7 should be improved.
Response: The quality of the SEM images in Figures 2, 6, and 7 has been improved
Comment 8: In most of the paragraphs for results and discussion, the analysis and discussion should be strengthened, especially in the mechanism analysis. Some contrastive analysis between this study and other related literature should be added, and some references can be used: High-temperature behavior of polyvinyl alcohol fiber-reinforced metakaolin/fly ash-based geopolymer mortar, Composites Part B - Engineering, 2022; Influencing factors analysis and optimized prediction model for rheology and flowability of nano-SiO2 and PVA fiber reinforced alkali-activated composites, Journal of Cleaner Production, 2022; Influence of SiO2 /Na2O molar ratio on mechanical properties and durability of metakaolin-fly ash blend alkali-activated sustainable mortar incorporating manufactured sand, Journal of Materials Research and Technology, 2022.
Response: The articles you suggested have been used as an object of comparison
Comment 9: The section of 4. Conclusion should be revised. The main conclusion of this study should be presented in three or four pieces.
Response: Conclusions have been revised
Round 2
Reviewer 3 Report
The manuscript can be accepted for publication.
Author Response
Dear reviewer,
Thank you for appreciating our manuscript